# Ultrasound-Assisted Preparation of Chitosan/Nano-Activated Carbon Composite Beads Aminated with (3-Aminopropyl)Triethoxysilane for Adsorption of Acetaminophen from Aqueous Solutions

**DOI:** 10.3390/polym11101701

**Published:** 2019-10-16

**Authors:** Mohammadtaghi Vakili, Parisa Amouzgar, Giovanni Cagnetta, Baozhen Wang, Xiaogang Guo, Amin Mojiri, Ehsan Zeimaran, Babak Salamatinia

**Affiliations:** 1Green Intelligence Environmental School, Yangtze Normal University, Chongqing 408100, China; 20170076@yznu.cn; 2Discipline of Chemical Engineering, School of Engineering, Monash University Jalan Lagoon Selatan, Bandar Sunway, Selangor 47500, Malaysia; parisa.amouzgar@gmail.com; 3State Key Joint Laboratory of Environment Simulation and Pollution Control, Beijing Key Laboratory for Emerging Organic Contaminants Control, School of Environment, Tsinghua University, Beijing 100084, China; gcagnetta@mail.tsinghua.edu.cn; 4College of Chemistry and Chemical Engineering, Yangtze Normal University, Chongqing 408100, China; guoxiaogang0528@126.com; 5Department of Civil and Environmental Engineering, Graduate School of Engineering, Hiroshima University, Higashihiroshima 739-8527, Japan; amin.mojiri@gmail.com; 6Department of Biomedical Engineering, Faculty of Engineering, University of Malaya, Kuala Lumpur 50603, Malaysia; ehsanzeimaran@gmail.com

**Keywords:** chitosan, nano-activated carbon, composite beads, APTES, acetaminophen adsorption

## Abstract

A composite chitosan/nano-activated carbon (CS-NAC) aminated by (3-aminopropyl)triethoxysilane (APTES) was prepared in the form of beads and applied for the removal of acetaminophen from aqueous solutions. NAC and APTES concentrations were optimized to obtain a suitable adsorbent structure for enhanced removal of the pharmaceutical. The aminated adsorbent (CS-NAC-APTES beads) prepared with 40% *w*/*w* NAC and 2% *v*/*v* APTES showed higher adsorption capacity (407.83 mg/g) than CS-NAC beads (278.4 mg/g). Brunauer–Emmett–Teller (BET) analysis demonstrated that the surface area of the CS-NAC-APTES beads was larger than that of CS-NAC beads (1.16 times). The adsorption process was well fitted by the Freundlich model (*R*^2^ > 0.95), suggesting a multilayer adsorption. The kinetic study also substantiated that the pseudo-second-order model (*R*^2^ > 0.98) was in better agreement with the experimental data. Finally, it was proved that the prepared beads can be recycled (by washing with NaOH solution) at least 5 times before detectable performance loss.

## 1. Introduction

In recent years, improved living standards have led to the production, consumption, and release of pharmaceuticals and personal care products (PPCPs) into the environment. PPCPs in the environment are considered to be emerging contaminants because of their adverse effects on human health and/or ecosystems [1]. Among pharmaceuticals, acetaminophen (ACT) is one of the most extensively consumed analgesic medicines for pain relief [2]. Because of its high rate of consumption, high water solubility, and low absorption uptake in the body, large amounts of ACT are released into water resources [3]. It has been reported that ACT at high concentrations (>1.8 μg/L) can be toxic to aquatic organisms [4]. Moreover, because it cannot be completely metabolized by the human body, extensive use of ACT can cause significant damage to living organisms [1]. Therefore, due to its potential risks and adverse effects on living organisms, the presence of ACT in the aquatic environment is a significant problem that needs to be addressed. Among the water treatment techniques for removal of organic compounds from aqueous streams, adsorption is considered to be one of the most rapid, simple, and effective methods, as it has high removal efficiency. The adsorption process is a low-cost (in terms of design and operation) physical process that can separate toxic contaminants and thus avert the generation of biologically active by-products [5,6]. Moreover, cheap adsorbents or highly engineered ones with outstanding properties are being developed for pollutant removal [7,8].

Chitosan (CS) and its derivatives have gained considerable attention for their ability to remove organic pollutants from wastewater because of their low cost, availability, abundance, biodegradability, biocompatibility, nontoxicity, and high adsorption capability [9,10]. However, they suffer from some drawbacks, such as insufficient mechanical properties, poor acid stability, low surface area, low porosity, and low thermal resistance, which limit the performance of this material in the adsorption process [5,11]. Therefore, in order to improve its adsorption performance, CS needs to be modified before utilization in the adsorption process.

Combining CS with other materials to prepare composite adsorbents is an effective way to overcome chitosan’s shortcomings and improve its adsorption performance [12]. Composite-based adsorbents integrate the unique properties of the constituent materials and reveal new and improved characteristics because of the synergistic effect among the components [13]. Carbonaceous materials (e.g., carbon nanotubes, carbon nanofibers, graphene, mesoporous carbon, etc.) are the most common adsorbents in wastewater treatment because of their particular properties, such as large surface area, high porosity, and high adsorption site number [14,15,16,17]. Among them, nano-activated carbons (NACs) have received notable attention for their remarkable potential as adsorbents for organic pollutants [18]. Specifically, NACs have a high surface-area-to-volume ratio and controlled pore size distribution, with a changeable surface chemistry. Such properties can allow for high adsorption capacity, rapid equilibrium rates, and effectiveness over a broad pH range.

In our previous works, combinations of CS and NACs demonstrated that synergistic effects and reductions of the drawbacks of both components can be achieved [19,20]. Specifically, the presence of CS in the structure of the composite could stabilize NAC and prevent carbon leaching. In addition, CS could affect the formation of a porous network in the composite adsorbent structure, which may be effective for the adsorption of ACT molecules [19,20]. In this study, in order to further improve the composite, CS-NAC was aminated using (3-aminopropyl)triethoxysilane (APTES) as a functionalization agent. To the best of our knowledge, such an aminated composite has never been reported in literature. Nevertheless, the functional groups derived by APTES, as well as the composite structure and surface area, might play an important role in the adsorption of pollutants [21]. In particular, the increase in the density and number of functional groups may enhance adsorption behavior by increasing the hydrogen bond number and, consequently, the affinity towards the adsorbate [22]. In this study, the effects of NAC and APTES concentrations on the preparation of an aminated CS-NAC composite were investigated to optimize the structural and physicochemical properties the adsorbent for ACT adsorption.

## 2. Materials and Methods

### 2.1. Materials

Medium-molecular-weight chitosan in powder form (75–85% deacetylated, molecular weight 190–310 kDa), APTES (99% purity), sodium hydroxide (NaOH), and ACT were all purchased from Sigma-Aldrich, Saint Louis, MO, USA. These materials were used as received without further purification. Nano-activated carbon was supplied by US Research Nanomaterials, Inc. (Houston, TX, USA) (super-nano-activated carbon, particle size of <100 nm, obtained from bamboo). Glacial acetic acid, acetonitrile (HPLC grade), and formic acid (98%) were purchased from Merck (Darmstadt, Germany).

### 2.2. Preparation of Adsorbent

CS-NAC composite beads were prepared through a simple dropwise method. In detail, the chitosan solution was initially prepared by dissolving 1.5 g of chitosan powder in 100 mL of distilled water, followed by addition of 3 mL of acetic acid and stirring for 5 h at room temperature (25 °C) [23]. Then, different concentrations of NAC (0.15, 0.3, 0.45, 0.6, and 0.75 g) were added to the chitosan solution and stirred magnetically at 500 rpm and room temperature for 60 min. The obtained solutions were then exposed to ultrasound irradiation to achieve a homogenous mixture. Sonication was performed using a 20 kHz ultrasonic generator (Qsonica, model Q700–220v, Newtown, CT, USA) equipped with a titanium probe transducer. The tip of the probe (1 cm diameter) was placed at 4 cm depth in the liquid. The corresponding samples were taken out after 6 min of ultrasonication at 60% amplitude with 6 s intervals between on and off pulses. During sonication, the beaker was placed in a water bath to maintain the temperature at 35 °C. This solution was then dropwise poured in a 500 mL solution of 2 mol/L NaOH while it was gently stirred for 24 h. The formed beads were washed several times with distilled water until the washing solution reached pH = 7 and then kept in distilled water for future use.

Preparation of CS-NAC-APTES beads was conducted according to our previously described methodology [24]. Initially, APTES solutions were prepared by pouring desired amounts of APTES (2, 3, 4, and 5 mL) into 200 mL of distilled water and stirring for 30 min at room temperature. Then, the prepared CS-NAC beads were poured into the APTES solutions. The obtained solutions were shaken using an incubator shaker (ZHWY-100D, Labwit, Shanghai, China) at 150 rpm and 50 °C for 5 h. Afterward, the final product (i.e., CS-NAC-APTES beads) were filtered and rinsed several times with distilled water to remove the unattached APTES residues. Finally, all the beads were dried by freezing using a ScanVac Coolsafe 110-4 freeze drier (Lynge, Denmark).

### 2.3. Characterization of Adsorbents

Field emission scanning electron microscopy (FESEM, Hitachi SU8010, Tokyo, Japan) was utilized to reveal the surface morphology of the samples. Functional groups on surfaces of the samples were investigated using Fourier transform infrared spectroscopy (FTIR, Thermo Scientific Nicolet IS10, Waltham, MA, USA). Zeta potential was measured with a Zetasizer Nano ZS (Malvern, Worcestershire, UK). The surface area was calculated on the basis of the Brunauer–Emmett–Teller (BET) method using a Micromeritics ASAP 2020 surface and porosity analyzer (Norcross, GA, USA).

### 2.4. Adsorption and Desorption Experiments

The prepared adsorbents were tested for adsorption of ACT from aqueous solutions in a series of batch adsorption experiments. Dried beads (50 mg) were mixed with 200 mL aqueous solutions containing different ACT concentrations. The initial pH of the ACT solution was adjusted by adding 0.10 mol/L of HCl or NaOH. Then, the solutions were shaken by an orbital shaker at 150 rpm for a certain amount of time. Process variables, including initial ACT concentration (10–200 mg/L) and contact time, were studied to determine the optimal condition for ACT removal. Moreover, adsorption kinetics and isothermal experiments were performed at different initial ACT concentrations (i.e., 10, 40, 80, 120, 160, and 200 ppm) at 25 °C for 24 h. After these adsorption experiments, 0.5 mL of the suspensions was withdrawn at intervals of 1 h (up to a maximum of 24 h), filtered, and analyzed using HPLC (Agilent 1200 Infinity Series, Santa Clara, CA, USA) equipped with a C18 column (Eclipse XDB-C18, 5 µm, 4.6 × 150 mm) for the accurate determination of residual ACT.

To assess the regenerability of the prepared adsorbents, adsorption experiments were conducted by mixing 50 mg of dried CS-NAC and CS-NAC-APTES beads with 200 mL of 100 mg/L ACT solution at pH 7 for 24 h. Then, the spent beads were separated, rinsed, and washed by 1.0 mol/L NaOH solution by agitation on an orbital shaker for 24 h. After desorption, the beads were repeatedly washed with deionized water to remove NaOH until a neutral pH was reached. The regenerated beads were reutilized for ACT adsorption experiments (as described above). Such adsorption cycles were performed five times.

## 3. Results

### 3.1. Effect of NAC Concentration

The ACT adsorptive efficiency of CS-NAC beads, prepared with different amounts of NAC, revealed that adsorption of ACT onto the CS-NAC beads was proportional to the amount of NAC (Figure 1a). An increase in NAC concentration from 10% *w*/*w* (0.15 g) to 40% *w*/*w* (0.6 g) enhanced the adsorption of ACT from 12.9% to 71.5%. Moreover, the beads synthetized with a higher NAC concentration attained equilibrium earlier than those containing a lower NAC percentage. Specifically, increasing the NAC concentration from 10% to 40% led to a decrease in the time necessary to reach equilibrium from 12 to 10 h, respectively. These results clearly show the beneficial effect of NAC on the composite adsorbent. Its role is likely to improve the porosity and surface area of the beads, making functional groups more accessible for adsorption of ACT [20].

A further increase in the NAC concentration up to 50% *w*/*w* (0.75 g) decreased the percentage of ACT removal to 49.1%. The presence of a high amount of NAC might cause a collapse of the adsorbent structure and also prevent the availability of adsorption sites in the CS matrix [25]. At a higher NAC concentration (i.e., low CS amount), the dropwise added CS-NAC solution in the NaOH solution could not solidify to form beads. This may be due to the low polymerization and entanglement rate of CS in solution caused by the presence of a low amount of CS [23]. Therefore, the CS-NAC beads prepared with a 40% *w*/*w* (0.6 g) NAC concentration, which showed the highest ACT uptake (71.5%), were chosen for the amination experiments.

### 3.2. Effect of APTES Concentration

APTES was applied as an amination agent to enhance the surface properties of the CS-NAC beads. Amination results showed that CS-NAC beads (0% APTES) actually had the lowest removal percentages (71.5%) (Figure 1b). The addition of APTES enhanced the ACT removal percentage, which could be attributed to the presence of more adsorption sites (NH_2_ groups) on the surface of the CS-NAC-APTES beads. In an aqueous environment, alkoxy groups of APTES are transformed into –OH groups and eventually form active Si–OH groups. Afterwards, the Si–OH groups can react with hydroxyl and carboxyl groups available on the CS-NAC surface. Subsequently, the number of amino groups within the structure of CS-NAC-APTES beads is increased [26]. At an APTES concentration of 2% *v*/*v*, ACT removal reached its maximum (97.3%), while a further increase in the APTES amount resulted in reduced ACT uptake. According to our previous work (Vakili et al. [24]), a high concentration of APTES in water for the amination process leads to APTES self-polymerization by the formation of Si–O–Si bonds. APTES polymeric particles can partially block pores in the bead structure and hinder active site accessibility. Moreover, APTES polymerization can stop additional bindings and further reactions with the surface of the adsorbent. These results suggest the existence of a threshold value for APTES concentration for bead amination. Concerning equilibrium time for ACT adsorption, amination determined its substantial decrease: when the concentration of APTES increased from 0% to 2% *v*/*v*, the adsorption equilibrium time decreased from 10 to 6 h. Such time shortening is possibly due to presence of a larger number of adsorption sites with higher accessibility.

### 3.3. Characterization

The surface analysis of the CS-NAC and CS-NAC-APTES beads illustrated that amination improved CS-NAC bead surface area, pore volume, and pore size by 13.9%, 2.8%, and 4.5%, respectively (Table 1). These enhancements were likely caused by expansion of CS through amination, which created larger pores in the structure. Specifically, due to penetration of APTES molecules throughout the pores, pressure was imposed to the bead structure, causing an expansion of pore size.

The surface morphology of CS-NAC beads, before and after amination, both showed numerous tiny and irregular multilayer pores (Figure 2). However, CS-NAC-APTES beads had a relatively rougher surface with less clogged, more visible, and larger pores, as compared with CS-NAC beads. Moreover, it was obvious that CS-NAC-APTES beads possessed a three-dimensional porous network. This could be attributed to the presence of APTES in the surface of CS-NAC-APTES beads, which allowed enhancement of adsorption properties (i.e., more sorption sites, larger pores, and contact area).

Figure 3a presents the FTIR spectra of the CS-NAC and CS-NAC-APTES beads. In both cases, the broad peak between 3000 and 3700 cm^−1^ was attributed to O–H and N–H stretching vibrations. The bands around 2940 cm^−1^ were ascribed to the bending vibration of CH_2_. In addition, the peaks at 1650, 1460, and 1370 cm^−1^ were due to primary, secondary, and tertiary amides, respectively [27]. The peaks at 1060 and 1020 cm^−1^ corresponded to the stretching vibration of C=O [28]. It was found that the FTIR spectrum of CS-NAC-APTES beads was very similar to that of CS-NAC beads. However, some peaks were slightly shifted and had higher intensities than those of CS-NAC beads. In particular, peaks at 3390, 2940, 1650, 1460, 1140, and 1030 cm^−1^ were shifted to 3420, 2920, 1660, 1450, 1150, and 1015 cm^−1^, respectively. The increase in the intensity of these peaks could have been due to the presence of APTES molecules on the structure of CS-NAC-APTES beads. In aminated beads, peaks at 995 and 850 cm^−1^ had stronger intensity, ascribed to Si–OH stretching and Si–O–Si symmetric stretching, respectively [29]. Additionally, a new peak at 740 cm^−1^ appeared in the spectrum of CS-NAC-APTES. These changes confirmed the presence of APTES in the structure of the CS-NAC-APTES beads. In aqueous solution, APTES undergoes hydrolysis, and silane groups covalently bind hydroxyl and carboxyl groups of NAC [30]. Thus, the number of amino groups on the surface of APTES-CS-NAC beads would increase. Figure 3b illustrates the proposed molecular structure of the APTES-CS-NAC beads.

In adsorption processes, surface charge (zeta potential) often plays an important role in determining adsorbent performance by influencing the adsorption rate [31]. Zeta potential measurements of the CS-NAC and CS-NAC-APTES beads (as a function of pH) showed that it followed the same trend for both adsorbents and was relatively high at low pH values (Figure 4). However, the isoelectric point (IEP) of the CS-NAC-APTES beads (pH = 8) was found to be higher than that of CS-NAC beads (pH = 7). This could be due to the presence of a higher amount of NH_2_ groups the CS-NAC-APTES bead structure: loading by APTES molecules could make the surface charge of aminated beads more positive than that of the CS-NAC beads. At the IEP, the surface charge of an adsorbent is zero. When a solution’s pH value is lower or higher than the IEP, the adsorbent’s surface charge is positive or negative, respectively [32]. Obviously, at pH < IEP, positively charged adsorbent surfaces are more effective at binding anionic species. Here, CS-NAC and CS-NAC-APTES beads were positively charged at pH < 7 and pH < 8, respectively. At these pH values, ACT had a negative charge because of its lower IEP compared with the adsorbent surfaces, thus favoring its uptake on beads.

### 3.4. Adsorption Kinetics

An adsorption kinetics study was performed to investigate the adsorption mechanism of ACT on the CS-NAC and CS-NAC-APTES beads. Obtained results were fitted to pseudo-first-order (Equation (1)) and pseudo-second-order (Equation (2)) models:(1)$$q_{t}=q_{e}\left(1-e^{-\mathrm{kt}}\right)$$
(2)$$q_{t}=\frac{q_{e}v_{0}t}{q_{e+}v_{0}t}$$
where $q_{t}$ and $q_{e}$ represent the amount of adsorbed ACT at time t and equilibrium (mg/g), respectively, and k (1/min) and $v_{0}$ (g/mg/min) are the pseudo-first- and second-order rate constants, respectively [33].

In both adsorption systems, ACT adsorption capacities gradually increased with time. The times for equilibrium adsorption were found to be 10 and 14 h for CS-NAC and CS-NAC-APTES beads, respectively (Figure 5). Moreover, the equilibrium adsorption capacity of CS-NAC-APTES beads (210.1 mg/g) was higher than that of CS-NAC beads (165.78 mg/g) because of their enhanced adsorption properties. Table 2 shows the calculated kinetic parameters of the CS-NAC and CS-NAC-APTES beads. In both cases, the pseudo-second-order model was fitted well to the adsorption data. Correlation coefficient (*R*^2^) and chi-square (X^2^) values of the pseudo-second-order model were higher and lower, respectively, than those of the pseudo-first-order model. Additionally, the adsorption capacities of CS-NAC and CS-NAC-APTES beads calculated by the pseudo-second-order model were closer to experimental adsorption capacities. This implies that adsorption of ACT onto the prepared adsorbent could be a chemisorption process.

### 3.5. Adsorption Isotherm

An isotherm study can describe the interaction between adsorbent and adsorbate under constant conditions. The obtained results were analyzed using Langmuir (Equation (3)) and Freundlich (Equation (4)) isotherm models:(3)$$q_{e}=\frac{q_{m}C_{e}}{\frac{1}{b}+C_{e}}$$
(4)qe=KCe1n
where $q_{e}$ (mg/g) is the maximum adsorption capacity, $C_{e}$(mg/L) is the equilibrium ACT concentration, $q_{m}$ (mg/g) is the maximum adsorption capacity of ACT on adsorbents, and K and *n* are the Freundlich constants indicating adsorption capacity (mg/g) and the intensity of the adsorption, respectively [34].

At lower concentrations of ACT, the curves had steeper slopes (Figure 6), implying that an increase in adsorbate concentration results in an increase in the adsorption capacities of both adsorbents. A comparison of the fitting parameters of both models (Table 3) revealed that the Freundlich model fitted the adsorption data better than the Langmuir one, as indicated by the higher *R*^2^ and lower X^2^ values. This suggests that ACT uptake is a multilayer adsorption process that happens on a heterogeneous surface. Moreover, *n* values for both were found to be 1.7, indicating that adsorption of ACT is a favorable process [35].

### 3.6. Adsorption Mechanism

Experimental and analytical outcomes indicated that the prepared beads possessed heterogeneous surfaces as well as high hydrophilicity, porosity, and surface area, which are favorable for the adsorption of contaminants. The presence of a large number of functional groups (NH_2_ and OH) imparted polar characteristics to the surface chemistry of CS-NAC-APTES beads through physicochemical interactions with adsorbate molecules in aqueous solutions [36]. The mechanism of ACT adsorption likely follows three steps: (i) creation of hydrogen bonds between the surface functional groups and the OH group of the ACT, (ii) formation of an electron donor–acceptor complex, and (iii) electrostatic interactions [37]. Among them, the dispersion interactions between the electrons of the ACT’s aromatic ring and carbon layers on the CS-NAC-APTES beads could be the main reason for adsorption. Moreover, the presence of a large number of functional groups on the adsorbent, caused by the addition of APTES, significantly improved the dispersion interactions and increased the hydrogen bindings. Figure 3a shows that after adsorption of ACT, intensities of peaks at 3420, 2920, 1380, and 1060 cm^−1^ decreased and shifted to 3300, 2910, 1370, and 1020 cm^−1^, respectively. These changes might have been due to the interactions of ACT molecules with the functional groups on the CS-NAC-APTES beads.

### 3.7. Reusability

ACT adsorption capacities of the CS-NAC and CS-NAC-APTES beads in the consecutive five regeneration cycles had similar trends, with a gradual performance decrease (Figure 7). Such a reduction in the adsorption capacities of both adsorbents could be due to saturation of functional groups and incomplete desorption [38]. However, after five cycles, the adsorption capacities of CS-NAC and CS-NAC-APTES beads were lowered to 189.12 mg/g (9.9% decrease) and 131.2 mg/g (20.8% decrease). Such a difference in adsorption capacity loss might be attributed to the improved adsorption performance of the aminated CS-NAC-APTES beads because of their higher number of functional groups on the bead surface. Hence, CS-NAC-APTES is also superior to CS-NAC in terms of reusability.

## 4. Conclusions

Aminated CS-NAC beads were successfully prepared for effective elimination of ACT from aqueous solutions. The highest enhancement in the ACT adsorption capacity of CS-NAC-APTES beads was obtained using 40% *w*/*w* NAC and 2% *v*/*v* APTES. The maximum monolayer adsorption capacities of ACT onto CS-NAC and CS-NAC-APTES beads were found to be 278.4 and 407.83 mg/g, respectively. The kinetic and isotherm experiments showed that pseudo-second-order and Freundlich isotherm models were in good agreement with the experimental results. These two models suggest that ACT adsorption is governed by chemisorption, occurring on multilayer heterogeneous surfaces. The regeneration experiment suggests that both materials have good regenerability and can be applied in the adsorption process for at least five cycles, although CS-NAC-APTES showed a less marked decrease in performance. Thus, CS-NAC-APTES beads are suitable adsorbents for the adsorption of ACT from wastewater.

## Figures and Tables

**Figure 1 polymers-11-01701-f001:**
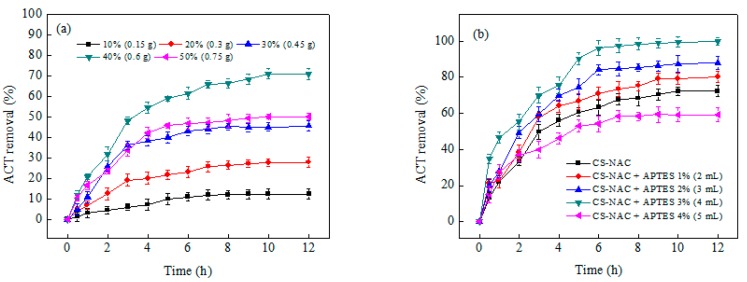
Effect of (**a**) nano-activated carbon (NAC) concentration on (3-aminopropyl)triethoxysilane (ACT) removal percentage using chitosan (CS)-NAC beads, and (**b**) APTES concentration on ACT removal percentage using CS-NAC-APTES beads (50 mg of beads in 200 mL of 10 mg/L ACT solution at pH 7 for 12 h).

**Figure 2 polymers-11-01701-f002:**
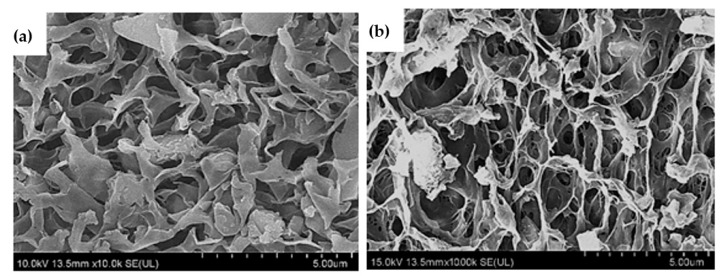
Field emission scanning electron microscopy (FESEM) images of (**a**) CS-NAC and (**b**) CS-NAC-APTES beads.

**Figure 3 polymers-11-01701-f003:**
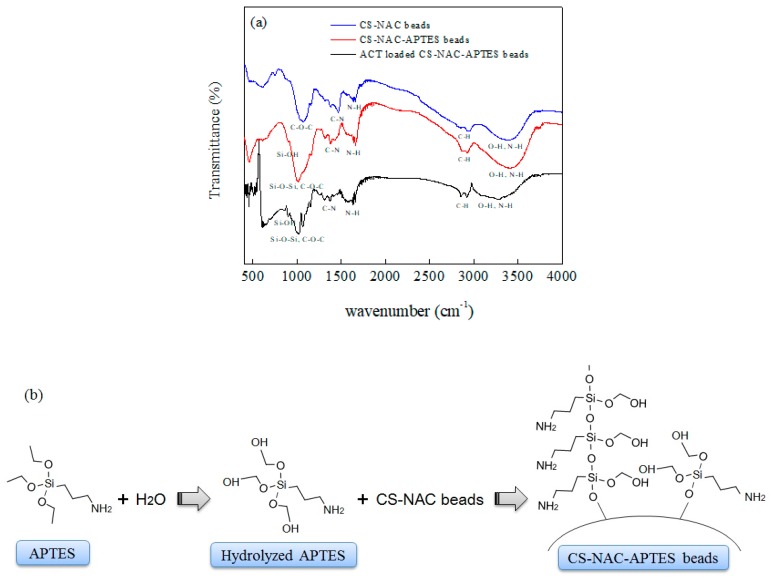
(**a**) Fourier transform infrared spectroscopy (FTIR) spectra of CS-NAC beads, CS-NAC-APTES beads, and CS-NAC-APTES beads after ACT adsorption. (**b**) Schematic diagram for the amination of CS-NAC beads by APTES.

**Figure 4 polymers-11-01701-f004:**
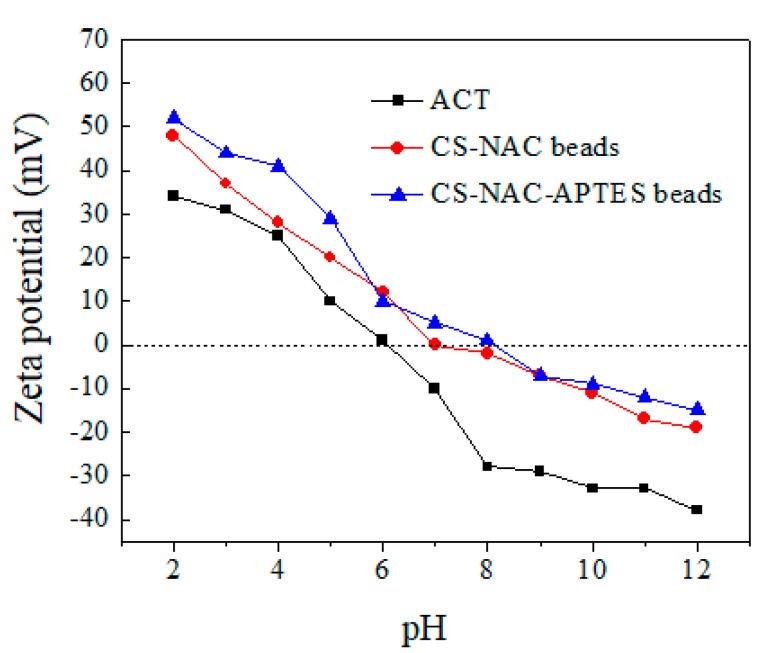
Zeta potential measurements for ACT, CS-NAC beads, and CS-NAC-APTES beads.

**Figure 5 polymers-11-01701-f005:**
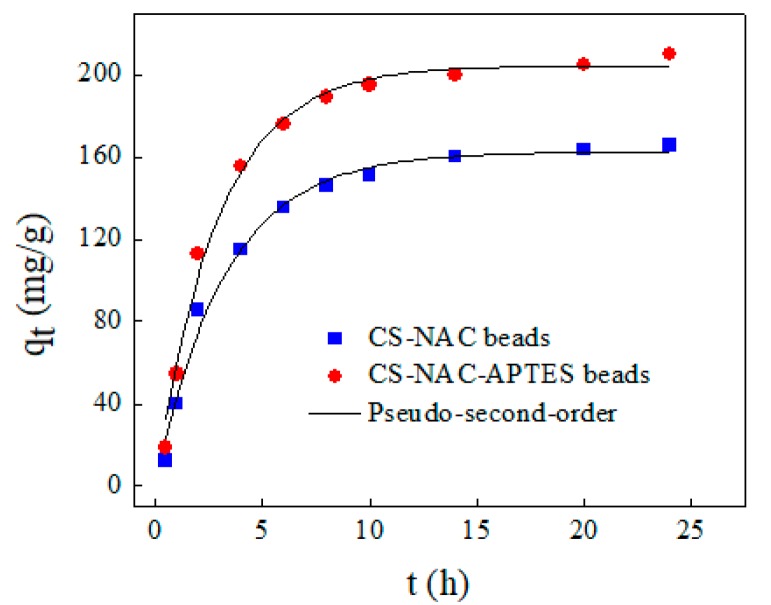
Adsorption kinetics of ACT onto CS-NAC and CS-NAC-APTES beads (50 mg of beads in 200 mL of 200 mg/L ACT solution at pH 7 for 24 h).

**Figure 6 polymers-11-01701-f006:**
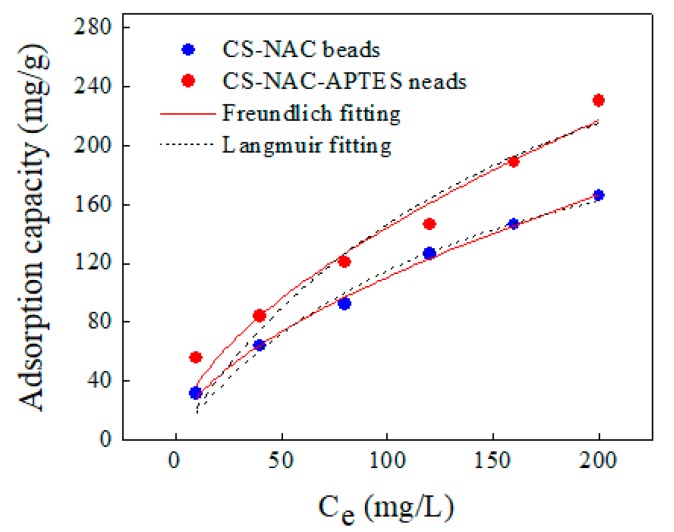
Adsorption isotherms of ACT onto CS-NAC and CS-NAC-APTES beads (50 mg of beads in 200 mL of ACT solution at pH 7 for 24 h).

**Figure 7 polymers-11-01701-f007:**
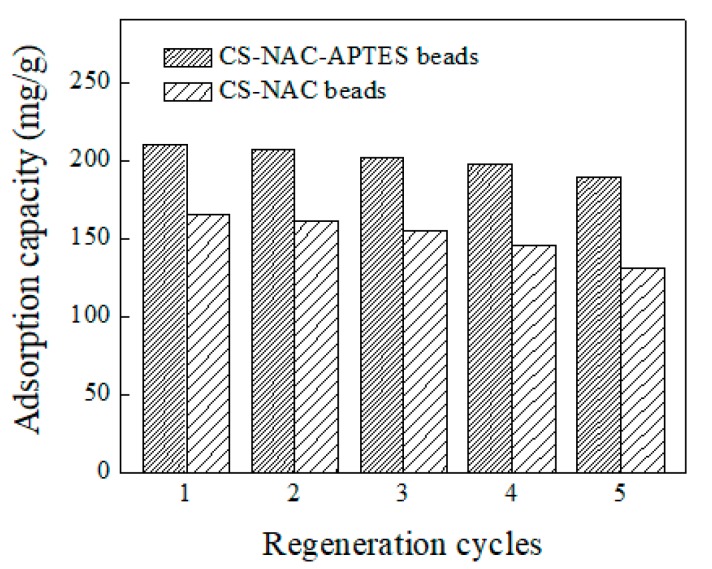
Effect of regeneration cycles on the adsorption capacity of CS-NAC and CS-NAC-APTES beads.

**Table 1 polymers-11-01701-t001:** Brunauer–Emmett–Teller (BET) characteristics of CS-NAC and CS-NAC-APTES beads.

Sample	Surface Area (m^2^/g)	Pore Volume (cm^3^/g)	Pore Size (nm)
CS-NAC beads	70.91	0.070	6.64
CS-NAC-APTES beads	82.32	0.072	6.96

**Table 2 polymers-11-01701-t002:** Characteristic parameters of the kinetic models.

Kinetic Model	Adsorbent	
	CS-NAC Beads	CS-NAC-APTES Beads
Pseudo-First-Order Model		
C_0_ (mg/L)	200	200
q_exp_ (mg/g)	165.78	210.19
q_cal_ (mg/g)	192.09	239.5
k (1/min)	0.307	0.344
$R^{2}$	0.977	0.973
$X^{2}$	65.44	123.3
**Pseudo-Second-Order Model**		
C_0_ (mg/L)	200	200
q_exp_ (mg/g)	165.78	210.19
q_cal_ (mg/g)	162.5	204.5
$v_{0}\;$(mg/mg/h)	66.89	94.78
$R^{2}$	0.988	0.988
$X^{2}$	35.9	51.175

**Table 3 polymers-11-01701-t003:** Characteristic parameters of the isotherm models.

Adsorbent	Langmuir				Freundlich			
	$q_{m}\;(\mathrm{mg}/g)$	b (L/mg)	$R^{2}$	$X^{2}$	KF (mg/g)	n	$R^{2}$	$X^{2}$
CS-NAC beads	278.4	142.5	0.976	62.88	7.36	1.69	0.996	10.62
CS-NAC-APTES beads	407.83	177.99	0.894	448.83	9.6	1.7	0.956	186.28

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
