# Peer review of "Ultrasound-Assisted Preparation of Chitosan/Nano-Activated Carbon Composite Beads Aminated with (3-Aminopropyl)Triethoxysilane for Adsorption of Acetaminophen from Aqueous Solutions"

_polymers, 2019, doi:10.3390/polym11101701_

Round 1

Reviewer 1 Report

Aminated adsorbents (CS-NAC-APTES beads) were prepared and used for the adsorption of acetaminophen, which deserves to be accepted only after some major revisions:
(1) Why acetaminophen was applied as the  adsorbate? There are so many other contaminants in water.
(2) The preparation of aminated CS-NAC beads should be briefly described in the manuscript, not just be cited.
(3) Why CS-NAC-APTES beads were prepared at 50 °C for 5 h? How about at other temperature for various reaction time?
(4) CS-NAC-APTES beads were rinsed several times, what are the solutions used for washing?
(5) BET surface areas and pore sizes of the samples should be given.
(6) To reveal the adsorption mechanism, XPS analysis of the samples before and after adsorption of acetaminophen should be given.
(7) The formation of porous nano carbon-based networks such as 3D GO-based porous composites has been reported previously, some important references should be cited in the manuscript in Introduction section: Journal of colloid and interface science, 2018, 532, 58-67; Ecotoxicology and environmental safety, 2019, 176, 11-19;  Journal of hazardous materials, 2019, 369, 214-225; etc.
(8) From SEM images, did the samples possess 3D structures?

Reviewer 2 Report

see attached file.

Round 2

Reviewer 1 Report

It can be accepted.

Reviewer 2 Report

I have no further comments.